# Effects of Short-Term Dietary Protein Restriction on Blood Amino Acid Levels in Young Men

**DOI:** 10.3390/nu12082195

**Published:** 2020-07-23

**Authors:** Kim A. Sjøberg, Dieter Schmoll, Matthew D. W. Piper, Bente Kiens, Adam J. Rose

**Affiliations:** 1Section of Molecular Physiology, Department of Nutrition, Exercise and Sports, Faculty of Science, University of Copenhagen, 2100 Copenhagen, Denmark; kasjoberg@nexs.ku.dk (K.A.S.); bkiens@nexs.ku.dk (B.K.); 2Sanofi-Aventis Deutschland GmbH, Industriepark Hoechst, 65926 Frankfurt am Main, Germany; Dieter.Schmoll@sanofi.com; 3School of Biological Sciences, Monash University, Clayton, VIC 3800, Australia; Matthew.Piper@monash.edu; 4Nutrient Metabolism & Signalling Laboratory, Department of Biochemistry and Molecular Biology, Metabolism, Diabetes and Obesity Program, Biomedicine Discovery Institute, Monash University, Clayton 3800, Australia

**Keywords:** amino acids, dietary protein, meal feeding, fasting, restriction

## Abstract

Pre-clinical studies show that dietary protein restriction (DPR) improves healthspan and retards many age-related diseases such as type 2 diabetes. While mouse studies have shown that restriction of certain essential amino acids is required for this response, less is known about which amino acids are affected by DPR in humans. Here, using a within-subjects diet design, we examined the effects of dietary protein restriction in the fasted state, as well as acutely after meal feeding, on blood plasma amino acid levels. While very few amino acids were affected by DPR in the fasted state, several proteinogenic AAs such as isoleucine, leucine, lysine, phenylalanine, threonine, tyrosine, and valine were lower in the meal-fed state with DPR. In addition, the non-proteinogenic AAs such as 1- and 3-methyl-histidine were also lower with meal feeding during DPR. Lastly, using in silico predictions of the most limiting essential AAs compared with human exome AA usage, we demonstrate that leucine, methionine, and threonine are potentially the most limiting essential AAs with DPR. In summary, acute meal feeding allows more accurate determination of which AAs are affected by dietary interventions, with most essential AAs lowered by DPR.

## 1. Introduction

Of the dietary macronutrients, protein is pivotal, as the digestion of protein yields amino acids which are vital for life [1,2]. While a severe restriction or a lack of protein is not compatible with life, dietary protein restriction to a certain threshold seems to be beneficial [3,4,5]. Indeed, dietary protein intake rates are positively related to type 2 diabetes risk [6] as well as all-cause mortality [7] in humans. Furthermore, several pre-clinical studies have demonstrated that dietary protein restriction (DPR) retards several age-related diseases such as type 2 diabetes, Alzheimer’s disease, and perhaps even certain cancers [8,9,10,11,12,13,14]. Importantly, several human trials have shown that the basic responses to DPR are conserved from mice to humans [10,13,15].

We and others have shown that the dietary restriction of amino acids can mimic the effects of dietary protein restriction [13,16,17]. Indeed, we recently demonstrated that the restriction of certain amino acids such as threonine and tryptophan was sufficient and necessary to confer the effects of DPR using casein-based diet feeding studies of mice [17]. However, far less is known about which AAs are necessary for the effects of DPR in humans. Thus, to learn more about the effects of DPR in humans, in a pilot study using a within-subjects diet design, we examined the effects of dietary protein restriction on fasting- and meal feeding-induced changes in blood plasma amino acid levels.

## 2. Materials and Methods

### 2.1. Human Study Design

For this pilot study, five healthy, lean male volunteers, aged 25.6 ± 0.4 years, with a body weight of 75.9 ± 5.3 kg (mean ± SEM), participated. Subjects consumed a controlled diet low in protein (~9%EP; ~12.8 MJ/d) for 7 days followed by a wash-out period for 7 days on their habitual, mixed diet (~20%EP; ~9.4 MJ/d) and protein intake averaged 0.94 ± 0.03 and 1.57 ± 0.22 g/kg/d in the PR and habitual diet, respectively (Table 1). Blood samples were obtained from the antecubital vein at selected time points in the morning after an overnight fast. A meal test was conducted before as well as at d7 of the experiment, whereby subjects ate an eucaloric meal (i.e., 60 kJ/kg BM containing 15%E or 9%E protein for normal-protein diet (NPD) vs. low-protein diet (LPD), respectively) with venous blood samples taken before as well as at selected times after the meal consumption. All subjects gave informed consent prior to their participation. This study was approved by the Copenhagen Ethics Committee (H-3-2012-129) and was executed in accordance with the code of ethics of the World Medical Association (Helsinki II declaration). Some results of this study have been published previously [10].

### 2.2. Blood Plasma Amino Acid Profiling

Blood plasma amino acid profiling was conducted using LC–MS/MS. Analysis, including sample preparation and derivatization, was performed using the EZ:faast™ kit (Phenomenex, Torrance, CA, USA) according to the user’s manual. For LC–MS/MS detection, an electrospray ionization-triple quadrupole mass spectrometer (Quantum Ultra, Thermo, Waltham, MA, USA) coupled to a liquid chromatography system (UltiMate3000, Dionex, Sunnyvale, CA, USA) controlled by Xcalibur 2.0.7 software with Dionex Chrom MS link 6.80 was used. Chromatographic separation was achieved on a 250 × 2 mm EZ:faast AAA–MS column (Phenomenex) at 35 °C with a flow rate of 250 μL/min; the autosampler temperature was set to 10 °C. A sample volume of 1 μL was injected onto the column. Eluents consisted of 10 mM ammonium formate in water (A) and 10 mM ammonium formate in methanol (B). Initial conditions (0 min) were 68% B, then a linear gradient was applied within 13 min to 83% B. The system returned to initial conditions within 4 min and equilibrated for 7 min, resulting in a total run time of 23 min per sample. The column flow was directly converted into the H-electrospray ionization (HESI) source of the mass spectrometer, which was operated in the positive ion mode. Capillary and vaporizer temperatures were maintained at 350 and 50 °C, respectively. Sheath gas and auxiliary gas were operated at 40 and 25 (pressure, arbitrary units), and no ion sweep gas was applied. Collision energy and tube lens offset were adjusted accordingly to obtain the highest response for the specific amino acid. Quantification was performed via peak area ratios applied to internal standards provided in the kit.

### 2.3. In Silico Analyses

Dietary amino acid supply from the low-protein diet was calculated according to Dankost 3000 (Dankost ApS, Copenhagen, Denmark) and compared against the amino acid distribution in the human exome, as previously described [18]. Exome matching used the human proteome sequences downloaded from Ensembl in 2018 (GRCh38 assembly). Briefly, to find the average amino acid composition of the exome, we find the relative proportion of each amino acid for each protein. We then compute the average relative proportion of each amino acid across all proteins in the proteome, thus generating an amino acid ratio representative of every gene in the exome.

### 2.4. Statistical Analysis

Statistical analyses were performed using one- or two-way analysis of variance (ANOVA) with t repeated measures, with Holm–Sidak-adjusted post-tests. All analyses were carried out with SigmaPlot 14 (Systat Software, Inc., Santa Clara, CA, USA) and data were visualized with GraphPad Prism 8.0 (GraphPad Software, Inc., San Diego, CA, USA). Statistical details can be found within the Figure legends. Differences between groups were considered significant when *p* < 0.05.

## 3. Results

### 3.1. Effects Of Dietary Protein Restriction on Fasting Venous Blood Plasma Amino Acid Levels

A schematic of the study design is shown (Figure 1A). Initially, we examined the effects of dietary protein restriction on blood amino acid concentrations during fasting. Non-essential (NEAAs; Figure 1B–K) and essential (EAAs; Figure 1L–T) amino acids are shown. Of the NEAAs, only TYR (Figure 1K) was significantly lower in the fasted state with LPD. Of the EAAs, only LEU (Figure 1N) and PHE (Figure 1Q) were altered with fasted levels lower 7 d after LPD feeding in the fasted state, which were then no different to initial levels 7 d after a NPD feeding.

In addition to the proteinogenic AAs, non-proteinogenic AAs were also measured. Of these, none were affected by the diet feeding in the fasted state (Figure 2).

### 3.2. Effects of Dietary Protein Restriction on Venous Blood Plasma Amino Acid Levels with Isocaloric Meal Feeding

We also conducted isocaloric meal feeding studies before as well as at the end of the 7 d LPD feeding period, which involved the subjects ingesting a mixed meal of 60 kJ/kg BM containing 15% or 9% protein for NPD vs. LPD, respectively, after an overnight fast, with blood samples taken before as well as at selected intervals after feeding. With regard to NEAAs (Figure 3), only TYR (Figure 3I) was affected, both in the fasted state as well as at 60–240 min after meal feeding. With regard to EAAs, all three branched chain amino acids including ILE (Figure 4B), LEU (Figure 4C), and VAL (Figure 4I) were affected, with levels being lower after meal feeding. In addition, the aromatic AA PHE (Figure 4F) was also only lower after meal feeding. Furthermore, there was a main effect of diet during the meal feeding test for the EAA LYS (Figure 4D) and THR (Figure 4G).

In addition to the proteinogenic AAs, we also assessed the non-proteinogenic AA (Figure 5). Of these, only 1MHIS (Figure 5A) and 3MHIS (Figure 5B) were affected, with levels lower during the meal feeding.

### 3.3. Human AA Exome Comparision Indentifies Potentially Limiting Essential Amino Acids With DPR

Previously, it was shown that one can predict which AAs will be most limiting in a particular diet by exome AA abundance [18]. Hence, we examined this. For this, we calculated the intake of each AA from the LPD (Figure 6A) as well as the %AA abundance in the human exome (Figure 6B). A comparison of this demonstrated that certain AAs such as GLN/GLU, ASN/ASP, ARG and PHE are abundant in the LPD relative to % representation in the human AA exome and thus unlikely to be limiting (Figure 6C). However, some NEAAs such as CYS, GLY, ALA, and SER were potentially limiting. In addition, EAAs such as MET, THR, and LEU were also limiting, with TRP and LYS being close to limiting (Figure 6C).

## 4. Discussion

Multiple studies have shown that dietary protein restriction promotes metabolic health via restriction of dietary amino acids. In particular, we have recently shown that restriction of essential amino acids is both sufficient and necessary to confer the metabolic remodeling that occurs with dietary protein restriction (DPR) in mice [17], less information about which amino acids do so in humans. Here, we show that, particularly during meal feeding tests, the blood plasma levels of essential amino acids isoleucine, leucine, valine (i.e., branched chain AA), lysine, phenylalanine, and threonine, as well as the non-essential AA tyrosine, were lower in young men undergoing DPR compared with their habitual diet. A similar signature was found in our former studies of mice [9,17], demonstrating mammalian conservation of these responses. Additionally, this AA signature is reminiscent of several studies that have shown higher blood levels of these amino acids as predictive of current or future type 2 diabetes risk in humans [19,20,21]. Given that dietary protein intake positively correlates with T2D risk [6,22], it is then possible that this signature is reflective of this association.

We have preciously demonstrated in mice, using casein-based diets, that the restriction of EAAs, particularly THR and TRP, was necessary and sufficient to confer the metabolic effects of DPR [17]. Thus, even though certain NEAAs were deemed to be the most limiting according to in silico exome matching (Figure 6), this is likely to be inconsequential [17]. LEU, MET and THR were shown to be the most limiting EAAs (Figure 6). However, of these, only the blood levels of LEU and THR were affected by low-protein feeding (Figure 4), and levels of LEU and MET can be metabolically spared in vivo by amino acids which share a similar metabolic pathway such as isoleucine and cysteine, respectively [13,23]. Thus, given that THR was the most limiting EAA identified (Figure 6) and its blood levels were lower with DPR (Figure 4), it is likely that the restriction of THR alone to levels within DPR would mimic the effects of DPR. Indeed, THR restriction without total AA restriction mimics the effects of DPR in mice, without the negative side-effects of reduced skeletal muscle mass [17]. In addition, we previously demonstrated that dietary THR restriction improves metabolic health in a pre-clinical mouse model of obesity driven type 2 diabetes [17]. Thus, future studies should closely examine the effects dietary threonine restriction in humans.

It is worth noting that the effects of DPR on certain blood plasma AA levels were only observed with the acute meal feeding test. This result is congruent with the observation that differences in biomarkers in response to dietary interventions are best observed with acute challenges such as feeding or fasting [24,25]. In our opinion, these acute nutritional interventions allow for bigger effect sizes, and less scatter, of biomarker levels within study groups. Thus, we conclude that it is advisable to conduct such nutritional ‘synchronization’ in dietary interventions.

There are several limitations of our work that should be noted. We used a small, homogenous population—young Danish men. Thus, whether similar responses would be found with humans of different age, sex and ethnicity warrants further investigation. The study design was a longitudinal fixed-order diet design, and thus future studies should consider using a randomized cross-over diet design. With the low-protein feeding, subjects consumed more energy overall, and particularly the consumption of both carbohydrates and fats also differed. Thus, we cannot formally rule out that the effects that we observe are directly related to the level of protein in the diet. Lastly, our subject number was rather low, and we acknowledge the possibility of false-positive/negative results due to the small sample size.

## 5. Conclusions

In summary, compared with measuring in the fasted state, acute meal feeding allows more accurate determination of which AAs are affected by dietary interventions, with most essential AAs lowered by DPR. In addition, those essential AAs predicted to be limiting by exome matching are indeed lower with DPR upon meal feeding, indicating that this method may be used to predict those essential AAs that are limiting in a particular diet.

## Figures and Tables

**Figure 1 nutrients-12-02195-f001:**
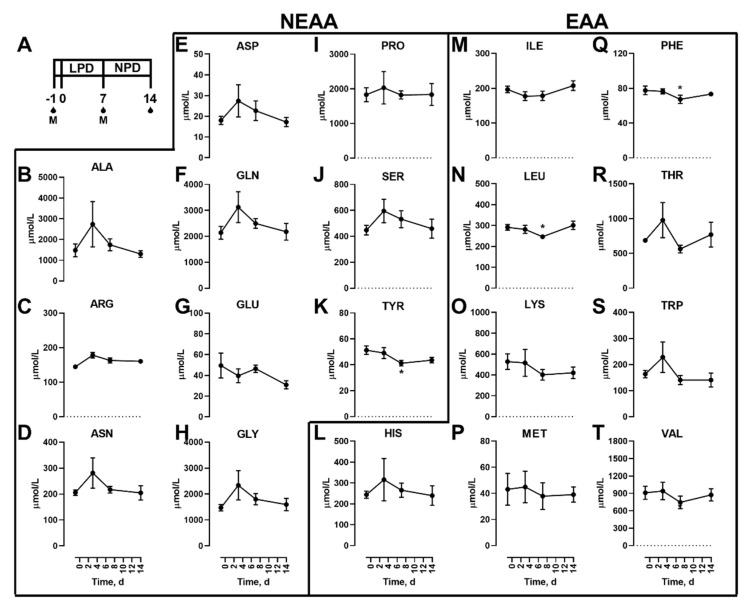
Effects of dietary protein restriction on blood serum proteinogenic amino acids during fasting. After an initial sample, young men (*n* = 5) underwent a 7 d period of consuming a protein-restricted diet, followed by a 7 d diet matching habitual protein intake (NPD), with blood samples taken in the overnight fasted state before, 3 d and 7 d after LPD, and at 7 d after NPD feeding. (**A**), schematic of the study design. The blood drops represent the fasting samples and M represents an isocaloric meal tolerance test. Blood serum amino acid levels including the non-essential amino acids (NEAAs) alanine (**B**), arginine (**C**), asparagine (**D**), aspartate (**E**), glutamine (**F**), glutamate (**G**), glycine (**H**), proline (**I**), serine (**J**), and tyrosine (K); and essential amino acids (EAAs) histidine (**L**), isoleucine (**M**), leucine (**N**), lysine (**O**), methionine (**P**), phenylalanine (**Q**), threonine (**R**), tryptophan (**S**) and valine (**T**) were measured. Data are the mean ± SEM. One-way RM ANOVA, different than time −1: * *p* < 0.05.

**Figure 2 nutrients-12-02195-f002:**
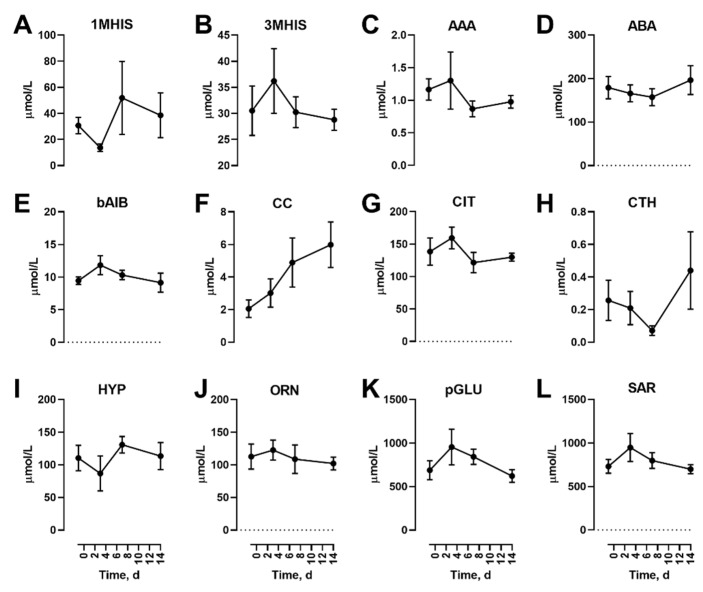
Effects of dietary protein restriction on blood serum non-proteinogenic amino acids during fasting. After an initial sample, young men (*n* = 5) underwent a 7 d period of consuming a protein-restricted diet, followed by a 7 d diet matching habitual protein intake (NPD) with blood samples taken in the overnight fasted state before, 3 d and 7 d after LPD, and at 7 d after NPD feeding. Blood serum amino acid levels including 1-methyl histidine (**A**), 3-methyl histidine (**B**), α-aminoadipate (**C**), α-aminobutyrate (**D**), ß-aminoisobutyrate (**E**), cystine (**F**), citrulline (**G**), cystathionine (**H**), hydroxyproline (**I**), ornithine (**J**), pyroglutamate (**K**) and sarcosine (**L**) were measured. Data are the mean ± SEM.

**Figure 3 nutrients-12-02195-f003:**
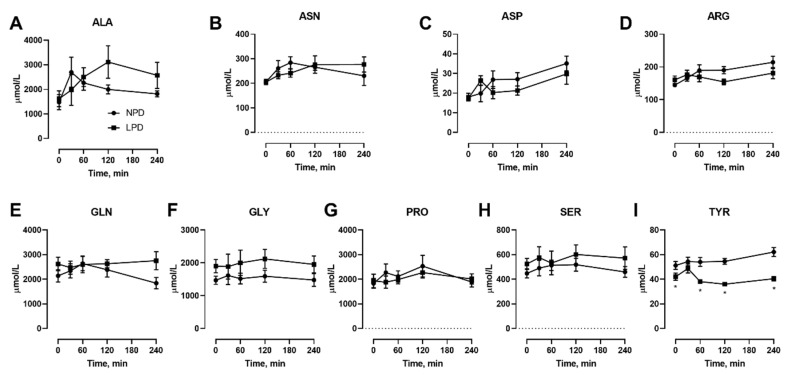
Effects of dietary protein restriction on blood serum non-essential amino acids during isocaloric meal feeding. Young men (*n* = 5) underwent a 7 d period of consuming a protein-restricted diet, with isocaloric meal tolerance tests conducted before and at the end of the 7 d period, with blood samples drawn at selected time points during the test. Blood serum amino acid levels including alanine (**A**), asparagine (**B**), aspartate (**C**), arginine (**D**), glutamine (**E**), glycine (**F**), proline (**G**), serine (**H**), and tyrosine (**I**) were measured. NPD: normal-protein diet; LPD: low-protein diet. Data are the mean ± SEM. Two-way RM ANOVA, different than NPD: * *p* < 0.05.

**Figure 4 nutrients-12-02195-f004:**
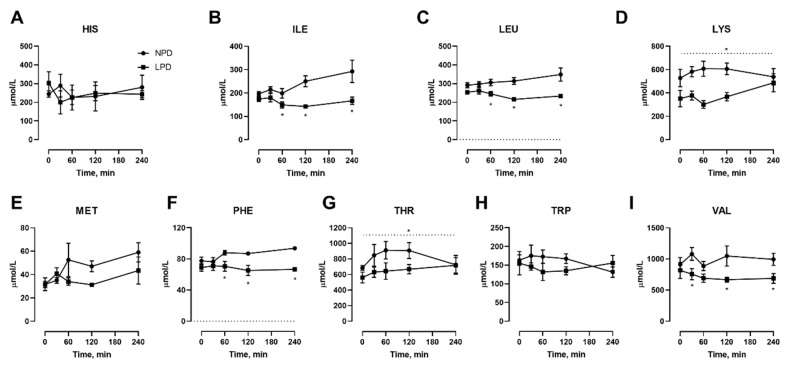
Effects of dietary protein restriction on blood serum essential amino acids during isocaloric meal feeding. Young men (*n* = 5) underwent a 7 d period of consuming a protein-restricted diet, with isocaloric meal tolerance tests conducted before and at the end of the 7 d period, with blood samples drawn at selected time points during the test. Blood serum amino acid levels including histidine (**A**), isoleucine (**B**), leucine (**C**), lysine (**D**), methionine (**E**), phenylalanine (**F**), threonine (**G**), tryptophan (**H**) and valine (**I**) were measured. NPD: normal-protein diet; LPD: low-protein diet. Data are the mean ± SEM. Two-way RM ANOVA, different than NPD: * *p* < 0.05. A dotted line over the entire data indicates a main effect of diet.

**Figure 5 nutrients-12-02195-f005:**
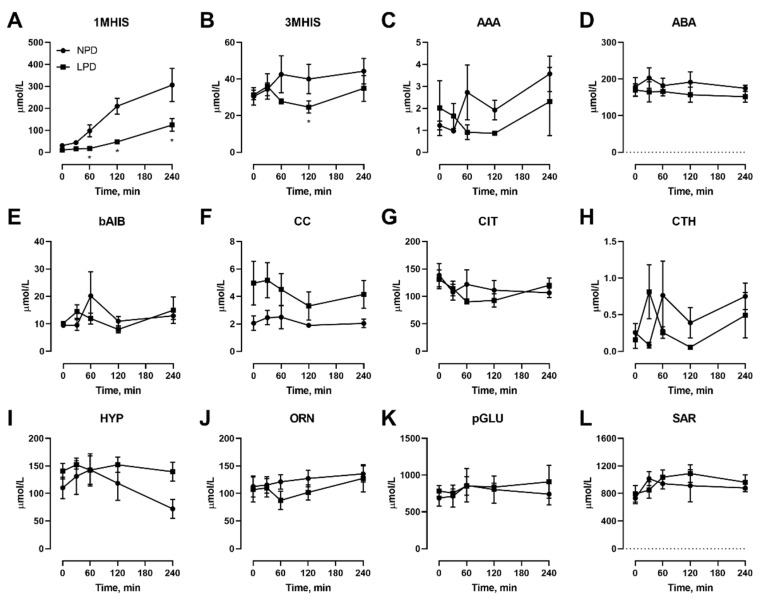
Effects of dietary protein restriction on blood serum non-proteinogenic amino acids during isocaloric meal feeding. Young men (*n* = 5) underwent a 7 d period of consuming a protein-restricted diet, with isocaloric meal tolerance tests conducted before and at the end of the 7 d period, with blood samples drawn at selected time points during the test. Blood serum amino acid levels including 1-methyl histidine (**A**), 3-methyl histidine (**B**), α-aminoadipate (**C**), α-aminobutyrate (**D**), β-aminoisobutyrate (**E**), cystine (**F**), citrulline (**G**), cystathionine (**H**), hydroxyproline (**I**), ornithine (**J**), pyroglutamate (**K**) and sarcosine (**L**) were measured. NPD: normal-protein diet; LPD: low-protein diet. Data are the mean ± SEM. Two-way RM ANOVA, different than NPD: * *p* < 0.05.

**Figure 6 nutrients-12-02195-f006:**
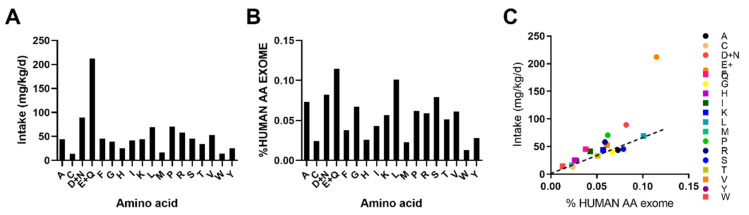
Identification of leucine, methionine, and threonine as potentially limiting essential amino acids with a low-protein intake. A: calculated daily proteinogenic amino acid intake rates during low-protein diet feeding. B: percent representation of AA abundance in the human exome. C: scatter plot of AA intake vs. %AA in the human exome. The dashed line dissects the center of THR, the most limiting EAA.

**Table 1 nutrients-12-02195-t001:** Diet compositions.

	Habitual Diet	Protein Restricted Diet
Energy, MJ/d	9.4 ± 0.8	12.8 ± 0.6
Alcohol	0 ± 0	0 ± 0
Protein, %E	20.2 ± 0.5	9.0 ± 0
CHO, %E	43.6 ± 0.6	71.0 ± 0
Fat, %E	36.1 ± 0.9	20.0 ± 0
Protein, g/d	111 ± 11	73.3 ± 3
Protein, g/kg BW/d	1.57 ± 0.22	0.97 ± 0.03

E: caloric energy; CHO: carbohydrate; BW: body weight.

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
