# Peer review of "Effects of Short-Term Dietary Protein Restriction on Blood Amino Acid Levels in Young Men"

_nutrients, 2020, doi:10.3390/nu12082195_

Round 1

Reviewer 1 Report

The authors investigated the effect of 7 days dietary protein restriction on blood amino acid levels. Generally, this is a well written and interesting manuscript. May major issue is a subject’s habitual diet that their habitual diet (more than 1.5 g protein /kg BW) is high protein rather than normal. Moreover, Energy intake is increased, while %fat energy is decreased in protein restricted diet. Please discuss these issues as a limitation or another strength. I think research novelty will remain alive if subject’s habitual diet is high protein diet.

The authors indirectly claimed that high protein diet is a risk of life style related diseases. The authors describe dietary protein restriction in the discussion. I think describing high protein diet (e.g. high protein vs. low protein/protein restriction) should be important and interesting.

It is not clear whether acute low protein meal (normal vs. low) affects blood AA kinetics or chronic (7 days) dietary protein restriction affects meal-induced blood AA kinetics.

The authors identified threonine as the most limiting EAA. Please describe the role of threonine, specifically how threonine affects energy metabolism and T2M/obesity.

Minor

L48: Please provide height or BMI

L133: Futhermore → Furthermore

Figure 6B, x-axis: %HUMANAA → %HUMAN AA

Author Response

The authors investigated the effect of 7 days dietary protein restriction on blood amino acid levels. Generally, this is a well written and interesting manuscript. May major issue is a subject’s habitual diet that their habitual diet (more than 1.5 g protein /kg BW) is high protein rather than normal. Moreover, Energy intake is increased, while %fat energy is decreased in protein restricted diet. Please discuss these issues as a limitation or another strength. I think research novelty will remain alive if subject’s habitual diet is high protein diet.

Response: This reviewer raises a good point here. This has now been discussed as a limitation within the discussion.

The authors indirectly claimed that high protein diet is a risk of life style related diseases. The authors describe dietary protein restriction in the discussion. I think describing high protein diet (e.g. high protein vs. low protein/protein restriction) should be important and interesting.

Response: Incorrect, we did not imply that a high-protein diet is a risk factor for life-style related illnesses. WE merely stated that these are correlated and perhaps this correlation could be explained by some individuals consuming a relatively low protein diet. The debate around dietary protein enrichment and health is complicated, and we feel that we do not have the reason nor scope within this manuscript to touch upon this topic and have decided to avoid this here as this should be done in a much more thorough way such as a detailed literature review.

It is not clear whether acute low protein meal (normal vs. low) affects blood AA kinetics or chronic (7 days) dietary protein restriction affects meal-induced blood AA kinetics.

Response: This is a good point and we agree this is not clear. An acute study where an initial meal test to start the low protein feeding trial could have been done to address this. Unfortunately, we did not do this. This has been discussed as a limitation.

The authors identified threonine as the most limiting EAA. Please describe the role of threonine, specifically how threonine affects energy metabolism and T2M/obesity.

Response: We have assessed this is mice (Yap et al. Nat Commun, 2020). We have added a discussion point on this.

Minor

L48: Please provide height or BMI.

Unfortunately data on height, and thus BMI, are not available

L133: Futhermore → Furthermore

Corrected

Figure 6B, x-axis: %HUMANAA → %HUMAN AA

Corrected

Reviewer 2 Report

This is an excellent short communication by Sjoberg et al., that helps bridge the gap between mouse and human research in DPD. I am generally quite supportive of this publication and think it will be of great interest to readers. There are few very minor changes to make that would elevate the quality of this manuscript:

1) There are minor spelling errors throughout--I noticed the words 'Prism' under 2.4, as well as 'after' under 3.2 were misspelled, for example.

2) The figures are very helpful and well laid out. Figure 6, the y axes could be spaced more appropriately to avoid cramped text.

3) In Panel 6C, there is a dashed line going through threonine, the 'most limiting AA.' I would suggest adding a line in down the center of the plot (representing a perfect correlation) as well? The authors can keep the threonine line if they wish, but personally I think it makes it quite hard to distinguish where the other AAs lie, and a star or bold outline might be more appropriate to designate threonine.

Author Response

This is an excellent short communication by Sjoberg et al., that helps bridge the gap between mouse and human research in DPD. I am generally quite supportive of this publication and think it will be of great interest to readers. There are few very minor changes to make that would elevate the quality of this manuscript:

There are minor spelling errors throughout--I noticed the words 'Prism' under 2.4, as well as 'after' under 3.2 were misspelled, for example.

Response: All spelling mistakes are now corrected.

The figures are very helpful and well laid out. Figure 6, the y axes could be spaced more appropriately to avoid cramped text.

Response: The x axes in Fig 6 A and B are now wider.

In Panel 6C, there is a dashed line going through threonine, the 'most limiting AA.' I would suggest adding a line in down the center of the plot (representing a perfect correlation) as well? The authors can keep the threonine line if they wish, but personally I think it makes it quite hard to distinguish where the other AAs lie, and a star or bold outline might be more appropriate to designate threonine.

Response: We understand and appreciate the reviewers point here. We drew the line through THR as we felt that this was the most appropriate given the message that we were trying to convey based upon our prior studies. Adding another line would simply be confusing and make the figure even more ‘busy’.